# Androgen-Regulated microRNAs (AndroMiRs) as Novel Players in Adipogenesis

**DOI:** 10.3390/ijms20225767

**Published:** 2019-11-16

**Authors:** Julia Jansen, Thomas Greither, Hermann M. Behre

**Affiliations:** Center for Reproductive Medicine and Andrology, Martin Luther University Halle-Wittenberg, 06120 Halle (Saale), Germany; julia.jansen@uk-halle.de (J.J.); thomas.greither@medizin.uni-halle.de (T.G.)

**Keywords:** androgens, adipogenesis, microRNA

## Abstract

The development, homeostasis, or increase of the adipose tissue is driven by the induction of the adipogenic differentiation (adipogenesis) of undifferentiated mesenchymal stem cells (MSCs). Adipogenesis can be inhibited by androgen stimulation of these MSCs resulting in the transcription initiation or repression of androgen receptor (AR) regulated genes. AR not only regulates the transcription of protein-coding genes but also the transcription of several non-coding microRNAs involved in the posttranscriptional gene regulation (herein designated as AndroMiRs). As microRNAs are largely involved in differentiation processes such as adipogenesis, the involvement of AndroMiRs in the androgen-mediated inhibition of adipogenesis is likely, however, not yet intensively studied. In this review, existing knowledge about adipogenesis-related microRNAs and AndroMiRs is summarized, and putative cross-links are drawn, which are still prone to experimental validation.

## 1. Introduction

Androgen regulation of gene transcription is mediated through testosterone, or the more bioactive derivate dihydrotestosterone (DHT), or any other androgenic hormone binding to the androgen receptor (AR), followed by intra-nuclear binding of the ligand-activated AR to androgen-responsive elements (ARE) in the promoter region of the respective gene and subsequent transcription initiation or repression by AR-recruited cofactors [1,2]. These genes do not necessarily have to be protein-coding, as the transcription of several non-coding genes has been shown to be regulated by the AR [3,4,5]. On a somatic level, several tissues including prostate, muscle, liver, breast, ovaries or fat are prone to androgen action [6,7,8,9]. With respect to adipose tissue, testosterone and DHT were demonstrated to inhibit the adipogenic differentiation (adipogenesis) of mouse and human mesenchymal stem cells [10,11,12,13,14].

Adipogenesis designates the differentiation process of a mesenchymal progenitor cell to a mature adipocyte. In the times of increasing obesity prevalence, it is of utmost importance to understand the process of adipogenic differentiation, as the excessive proliferation of the body’s fat mass is strongly associated with serious adverse conditions such as type 2 diabetes, mellitus, and cardiovascular diseases. The general developmental process of adipose tissue is composed of two components: (1) the increase of adipocyte numbers by differentiation (hyperplasia), and (2) the swelling of the single-cell by accumulation of triglycerides (hypertrophy). On the cellular level, at first mesenchymal stem cells (MSCs) are recruited through abrogation of the differentiation block (commitment phase), and subsequently differentiate to the mature adipocyte (terminal differentiation phase, see Figure 1).

On the molecular level, especially two master regulators of adipogenesis are essential for differentiation: the transcription factors peroxisome proliferator-activated receptor gamma (PPARγ) and CCAAT/enhancer-binding protein alpha (CEBPα), which both induce a plethora of different adipocyte-specific genes during the commitment phase of differentiation [15]. Furthermore, an essential signaling pathway for the induction of adipogenesis by PPARγ is the Wnt pathway [16]. Mesenchymal progenitor cells are kept in an undifferentiated state by the canonical Wnt pathway through the induction of cyclin-D1 and c-myc. This differentiation block is mediated through the direct binding of c-myc on the DNA sequence of PPARγ and C/EBPα following inhibition of transcription. By such mechanisms, canonical Wnt10b and Wnt1 are capable of inhibiting the expression of PPARγ and blocking adipocytic differentiation [17,18].

On the other hand, several signaling pathways like the MAPK pathway, the PI3K/Akt pathway, the cAMP/PKA/CREB pathway, and the TGF-β pathway play an adipogenesis-promoting role after induction of the differentiation program within the commitment phase [19]. In recent years, by the identification and analysis of microRNAs, an additional player for the regulation of these different signaling pathways on the post-transcription level has been unraveled. MicroRNAs are small (18–25 nt long), endogenous RNAs, that are non-coding but involved in the post-transcriptional silencing of gene expression by translation inhibition [20]. Also during adipocytic differentiation, a subset of microRNAs are differentially expressed and subsequently regulate the differentiation course by inducing a massive shift in the cellular phenotype by changes in the expression patterns of their downstream target genes.

Actually, adipogenesis-regulating microRNAs and androgen-regulated microRNAs (AndroMiRs) could still be seen as “two separated kingdoms”. However, by connecting adipogenesis-related microRNAs to those prone to androgen-regulation, interesting candidates for the molecular mechanisms of the well-known hypogonadism-induced fat tissue accumulation as well as potential therapeutic targets against this detrimental process might be identified. In this review, we aim to summarize the existing literature on both microRNA kingdoms and show potential connections between both of them.

## 2. MicroRNAs in Human Mesenchymal Stem Cell Adipogenesis

In the past years, over 30 microRNAs or microRNA families have been identified to be involved in the adipogenic differentiation process in human mesenchymal stem cells or preadipocytic cell lines (see Table 1). Furthermore, extensive studies on mouse cell lines or other species have revealed several more candidate microRNAs yet to be verified in human cells (see Appendix A).

Li et al. identified miR-10b as a critical regulator for balancing osteogenic and adipogenic differentiation of human adipose-derived stem cells (hADSCs) by repressing ‘mothers against decapentaplegic homolog 2’ (SMAD2) [21]. Its expression is negatively correlated to adipogenic markers like CEBPα, PPARγ and activating protein 2 (AP2). In 3T3-L1 cells, apolipoprotein 6 (Apol6), which acts as an oncogene in obesity-related cancers, was identified as further target of miR-10b-5p. Inhibition of miR-10b-5p encouraged the adipogenic differentiation of 3T3-L1 cells. However, it has no effect on cell proliferation [60]. Similarly, the overexpression of miR-27b blunts the induction of the two key regulators CEBPα and PPARγ and represses triglyceride accumulation at the late stages of adipogenic differentiation [26]. Kim et al. confirmed these results by demonstrating a similar action of miR-27a in the 3T3-L1 cell adipogenesis of obese mice [61]. Accordingly, Hu et al. showed in microarray analysis an increase of lipoproteinlipase (LPL) during adipogenic differentiation of hADSCs, while miR-27b is decreased [27]. In addition, miR-130a and miR-130b influence the PPARγ expression in human preadipocytes. QPCR arrays showed that miR-130a/b targeted both the mRNA coding region as well as the 3’untranslated region of PPARγ [38]. These effects were also replicated in mouse 3T3-L1 preadipocytes. Further microRNAs that inhibit the expression of PPARγ and C/EBPs and lead to an obstructed adipogenic differentiation in hMSCs are for example miR-155, miR-221 and miR-222 [43].

Yang et al. studied the effect of miR-1908 on the differentiation on hMADS. MiR-1908 is highly expressed in human multipotent adipose-derived stem cells (hMADS) and inhibits adipocyte differentiation by promoting the proliferation of hMADS cells and influencing the cell cycle through expanding the S phase and inhibiting the G1 phase [57]. Recently, studies with Simpson Golabi Behmel syndrome (SGBS) cells were conducted. For example, miR-107 inhibits adipogenic differentiation of SGBS cells via cell division protein kinase 6 (CDK6), which regulates neurogenic locus notch homolog protein 3 (Notch3) and his target ‘hairy and enhancer of spli-1’ (Hes1). Furthermore, miR-107 induces the attenuated triglyceride storage by impairing glucose uptake and triglyceride synthesis [34]. Adiponectin receptor 2 (ADIPOR2), a direct target of miR-375, has an inhibiting effect on the adipogenic differentiation of SBGS cells, too. Kraus et al. showed the inhibiting effect of androgens on adipogenic differentiation through an androgen receptor-mediated pathway [54]. Bork et al. performed investigations regarding miR-369-5p, which both impaired the proliferation of human MSCs and enhanced the accumulation of lipid droplets during adipogenic differentiation.

Accordingly, the expression of adiponectin (ADIPOQ) and ‘fatty acid-binding protein 4’ (FABP4) is reduced during differentiation after transfection with miR-369-5p [53]. MiR-149-3p inhibits adipogenic differentiation in BMSC by directly targeting ‘fat mass and obesity-associated protein’ (FTO). Knockdown of miR-149-3p led to a decreased expression of adipocyte-related genes including CEBPα, CEBPβ, CEBPδ, FABP4 and PPARγ, whereas osteogenic markers like alkaline phosphatase (ALP), ‘bone gamma-carboxyglutamic acid-containing protein’ (BGLAP), secreted phosphoprotein 1 (SPP1), collagen type 1 (COL1A1), and ‘bone morphogenetic protein 4’ (BMP4) increased. MiR-149-3p also acts as a regulator of the switch between adipogenic and osteogenic differentiation [42]. Similar effects were observed for miR-194 and its target gene COUP transcription factor II (COUP-TFII), which activates PPARγ expression. Enhanced expression of miR-194 leads to a reduced expression of COUP-TFII, whereas inhibition of the miR-194 expression leads to an increased COUP-TFII level [46]. In contrast, miR-17-5p and miR-106a could promote osteogenesis and decrease adipogenesis. However, miR-17-5p and miR-106a are directly targeting BMP2, which has a reverse effect on the differentiation of hADSC. Downregulation of BMP2 through RNA interference suppressed osteogenesis and increased adipogenic differentiation [22]. By the application of a dual-luciferase assay, Chen et al. identified RhoA and ‘mitogen-activated protein kinase 1’ (ERK1) as direct targets of miR-125a-3p and miR-483-5p. Downregulation of these microRNAs in hADSC resulted in a decreased ERK1/2 phosphorylation in the nucleus in subcutaneous adipose tissue of patients with multiple symmetric lipomatosis.

Furthermore, miR-125a-3p and miR-483-5p promote the *de novo* formation of adipose tissue in nude mice [36]. Mei et al. demonstrated that through regulating the ERK-MAPK pathway, the only active signaling during adipogenic, osteogenic and chondrogenic differentiation, miR-21 stimulates MSC differentiation on an early stage. In this context, the expression of the marker gene for adipogenesis, PPARγ, and the marker gene for osteogenesis, Cbfa-1, were both increased after transfection of MSC with miR-21 mimics, while miR-21 inhibition resulted in a reduced expression level of both genes [23]. MiR-26b knockdown inhibits the accumulation of lipid droplets in adipogenic differentiation in human preadipocytes. Furthermore, the expression levels of PPARγ, AP2, C/EBPα and hormone-sensitive lipase (HSL) are reduced in knockdown cells towards untreated cells [24]. Moreover, PTEN was identified as a direct target of miR-26a [25,57]. Zhang and colleagues identified tumor necrosis factor (TNFα) by use of bioinformatical methods as an indirect target of miR-29b. Via specificity protein 1 (SP-1), it acts as enhancer of the adipogenic differentiation. Thereby, lipid accumulation in hADSC is promoted and the mitotic clonal expansion is inhibited [29]. Further well-studied microRNAs, which promote adipogenic differentiation, are the miR-30 family. It directly targets ‘plasminogen activator inhibitor’ (PAI-1) and ‘anaplastic lymphoma kinase’ (ALK2) in hMADS and enforces adipocyte marker gene induction. Interestingly, only the silencing of both genes leads to a pro-adipogenic effect of miR-30c, while silencing of one target has no effect on adipogenesis [31]. Additionally, Zaragosi and colleagues identified miR-30 with the help of gain and loss of function studies as enhancer of adipogenesis. Via ‘Runt- related transcription factor’ (RUNX2), also known as Cbfa-1, miR-30 family acts as a key regulator balancing adipogenesis and osteogenesis [30]. The miR-320 family has a similar effect [49]. Recent studies investigated ‘lysine-specific demethylase 6A’ (KDM6A) as a target of miR-199a-3p which regulates WNT signaling downstream [47]. The promoting effect of mir-199a-3p could be validated in 3T3-L1 cells [62]. Wang et al. identified mir-342-3p as a further powerful promoter of the adipogenic differentiation. Both in humans and in obese mice, it is upregulated during adipogenesis. The inhibition of miR-342-3p results in a decreased expression of adipogenic markers like PPARγ, C/EBPα, FABP4, and LPL. By the use of luciferase assays, CtBP2 was confirmed as a direct target of miR-342-3p [51].

## 3. Androgen-Regulated microRNAs (AndroMiRs)

Although several tissues have been shown to be androgen-sensitive, the utmost studies performed on androgen-regulated microRNAs (AndroMiRs) are from prostate carcinoma (PCa) or breast carcinoma (BCa) and the respective cell lines, potentially resulting in a bias, as these model systems represent pathological tissues and therefore are not necessarily reflecting the miRNome of their corresponding somatic counterparts. With regard to the existing literature on identified AndroMiRs, this fact has to be considered. Thus, in many cases, studies on the androgen-regulation of these proposed AndroMiRs in somatic tissues are still warranted.

As the first AndroMiR, miR-125b was identified in the androgen-sensitive prostate carcinoma cell line LNCaP to be induced by the synthetic androgen R1881 treatment [63]. Furthermore, miR-125b stimulated androgen-independent growth of LNCaP cells, also by targeting of BCL2-antagonist (BAK1) [63]. MiR-125b expression in LNCaP cells was demonstrated to be significantly downregulated by treatment with the AR antagonist bicalutamid, and miR-125b also targeting the AR-repressor complex protein (NCOR2) [64]. An induction of miR-125b expression in LNCaP cells after stimulation with DHT was also observed [65]. Concordantly, Sun and colleagues reported AR-mediated regulation on the promoter of the miR-99a/let-7c/miR-125b-2 cluster host gene LINC00478, although showing repression of miR-125b in reaction to AR stimulation with R1881. Additionally, the expression of the miR-100/let-7a-2/miR-125b-1 cluster was not affected by androgen stimulation [66]. Also in the breast cancer cell line MDA-MB-453, treatment with the AR-agonist CI-4AS-1 resulted in a significant downregulation of miR-125b and miR-100 and induced the expression of their target gene metalloprotease-13 (MMP13) [67]. In a non-transformed cell system, Sen and colleagues demonstrated induction of miR-125b, but not miR-125a, expression in mouse granulosa cells upon stimulation with testosterone or dihydrotestosterone, while estradiol stimulation exhibited no effect [68].

As one of the most prominent oncogenic microRNAs, miR-21 was early identified to be upregulated by R1881 treatment in the androgen-sensitive PCa cell lines LNCaP and LAPC-4 [69]. Additionally, the same group intensely studied the miR-21 promoter region by bioinformatics and furthermore demonstrated AR recruitment to an ARE in this promoter region by chromatin immunoprecipitation (ChIP) [69,70]. Concordantly, Mishra and colleagues described the downregulation of miR-21 in AR-silenced prostate cancer cell lines (22Rv1 and MDA-PCA-2b), resulting in the increased expression of the miR-21 target gene TGFBR2, as well as the formation of a positive AR-miR-21 feedback loop in prostate epithelial cells [71]. Teng and colleagues observed the induction of miR-21 through AR activation by several dihydroepiandrosterone (DHEA) metabolites, among them DHT, by promoter recruitment visualized via ChIP in the hepatocellular carcinoma cell line HepG2 [72]. Contrarily, in the breast cancer cell lines MCF-7 and SK-BR-3 the AR was shown to downregulate miR-21 expression, primarily by recruiting HDAC3 to the miR-21 promoter [73].

Also in LNCaP, the expression of miR-101 was shown to be upregulated following R1881 treatment, while subsequently the expression of its target gene ‘enhancer of zeste homolog’ (Ezh2) was downregulated [74]. The same relationship could be demonstrated in mouse granulosa cells, where Ezh2 is initially inactivated via phosphorylation mediated by the PI3K/Akt pathway, then Ezh2 transcript is downregulated via the testosterone-mediated induction of miR-101 expression [75]. In the neuronal cell line SH-SY5Y and the glioblastoma cell line U251, miR-101 was upregulated after testosterone stimulation resulting in the downregulation of CYP2D6 [76]. Guo et al. verified an AR binding site in the promoter region of miR-101 via ChIP, thus linking AR-induced autophagy inhibition in prostate cancer cell lines to miR-101 upregulation [77].

MiR-221 was initially detected as regulator of Dvl2 and being upregulated in an androgen-insensitive LNCaP-AI cell line in comparison to the original LNCaP, while miR-222, miR-21, miR-125b, and miR-101 were also differentially expressed in these cells [78]. Gui and colleagues identified miR-221/222 as AR-repressed gene, showing downregulation of pri-miR-221/222 after androgen stimulated chromatin modification of the miR-221/222 host gene promoter region [79]. In line with these findings are the results of Sun et al., showing the involvement of miR-221/222 in the development of castration-resistant prostate carcinoma [80,81,82,83].

MiR-27a was identified as oncomiR in PCa by targeting the AR corepressor Prohibitin (pHB) [84]. Furthermore, for the miR-23a/27a/24-2 cluster it was demonstrated that the AR in PCa cell lines not only induces the transcription by binding to an ARE in the promoter region, but also accelerates the processing of the pri-miR-23a/27a/24-2 cluster [84]. Concordantly, Mo and colleagues showed the upregulation of miR-27a expression (besides miR-133b and miR-19a) by DHT stimulation of LNCaP cells, and proposed ‘ATP-binding cassette transporter’ (ABCA1) and ‘sister chromatid cohesion protein’ (PDS5B) as target genes of miR-27a [85]. In castration-resistant PCA, miR-27a was found to be repressed by the PI3K pathway, thereby levering the repression of its target gene MAP2K4 [86]. In the endothelial cell line EA.hy926 as well as in HUVEC, DHT stimulation downregulated miR-27a expression and upregulated expression of its target gene TFPIα [87]. Also, in a mouse model of the polycystic ovary syndrome characterized by androgen excess, DHT facilitates the upregulation of miR-27a in the granulosa cells resulting in a feedback loop by miR-27a targeting the transcription factor Creb1 [88].

MiR-32, as well as miR-148a, was demonstrated to be upregulated in LNCaP cells after stimulation with DHT and to target BTG2 [89]. By siRNA-mediated knockdown of AR in prostate cancer cell lines, miR-32 was found to be upregulated and to enhance NSE activity, thereby promoting neuroendocrine differentiation of the prostate cancer cells [90].

In a Northern blot approach, miR-200c was observed to be differentially expressed between androgen-sensitive and androgen-insensitive PCa cell lines [91]. Furthermore, members of the miR-200 family comprising miR-200a-c were found to be upregulated by R1881 treatment in AR-induced PC-3 cells, with the highest increase of miR-200b expression, resulting in the suppression of proliferation and invasiveness of the prostate carcinoma cells [92]. Furthermore, miR-200a and miR-200b are higher expressed in the androgen-sensitive LNCaP cell line in comparison to androgen-insensitive DU145, and siRNA-mediated silencing of AR in LNCaP leads to a decrease in miR-200a/b and an increase of their target gene ‘zinc finger E-box-binding homeobox 2’ (ZEB2) [93].

By luciferase reporter assays and ChIP analyses, the miR-1-2 promoter region was shown to be activated by the AR, and miR-1 targeting ‘tyrosine-protein kinase’ (SRC) in DU145 derived prostate cancer cell lines [94]. Interestingly, the miR-1-2 promoter is also targeted by ‘Kruppel like factor 4’ (Klf4), which expression itself is induced by the AR [95]. Additionally, androgen-induced miR-1 downregulates TCF7 in prostate cancer cell lines and therefore negatively impacts the Wnt signaling pathway [96]. Another target gene of miR-1, in the context of the transition of PCa from androgen-sensitive to castration-resistant, is ZBTB46 [97].

MiR-375 and miR-141 were initially detected to be upregulated in the serum of patients with castration-resistant prostate carcinoma [98]. In LNCaP cells, significant overexpression of miR-375 and miR-141 was shown after stimulation with DHT for 24 h [99]. Conversely, Chu and colleagues demonstrated low expression of this microRNA and hypermethylation of the miR-375 promoter in the AR-negative PCa cell lines Du145 and PC-3 [100]. The same effect was observed in differentiating SGBS preadipocytes, where testosterone or DHT stimulation led to the downregulation of miR-375 during the differentiation course and upregulation of its target gene ADIPOR2 [54].

Other AndroMiRs include let-7a, b, c [101], let-7d [102], miR-135a [103,104], miR-141 [99,105], members of the miR-17~92 cluster [106,107], miR-216a [108,109], miR-29 family members [110,111] and miR-30 family members [99,112]. Additional proposed androgen-regulated microRNAs—yet not intensively studied—are given in Table 2.

Interestingly, Panneerdoss and colleagues studied testosterone-specific microRNA signatures in mouse sertoli cells, identifying two X-linked microRNA clusters (Cluster 1: miR-743a, miR-471, miR-741, miR-463, miR-880, miR-878, and miR-871. Cluster 2: miR-201 and miR-547) highly induced by testosterone [126,127]. Although the AR is regulating the expression of many microRNAs, AR translation is also repressed by several microRNAs (an actual overview is given in Appendix A), therefore, adding an additional regulative layer to the interplay between these two factors in differentiation processes.

## 4. AndroMiRs Putatively Involved in Adipogenesis

Comparing the androgen-regulated microRNAs (AndroMiRs) to the adipogenesis-related microRNAs (AdipoMiRs), several overlapping candidates can be extracted from the existing literature. In Figure 2, putative candidates for microRNAs involved in the androgen-mediated inhibition of the adipogenesis are given.

MiR-17-5p/miR-106a was shown to determine lineage commitment in early hMSC differentiation, wherein the upregulation was of both microRNAs suppressed BMP2 expression and therefore shifting the MSC fate towards adipogenesis [22]. However, in PCa cell lines androgens triggered the upregulation of the miR-17-92 cluster [106,107], therefore, analysis of the regulation in somatic tissues remains crucial for the determination of the role of this cluster in differentiation processes. MiR-21 upregulation increased the potential of adipogenic and osteogenic differentiation of hMSCs via modulation of the ERK-MAPK4 pathway [23], therefore, being no adipogenesis-specific modulator. In line with this fact, the androgen-induced upregulation of miR-21 observed in several studies [69,70,71,72] may point towards a general role of this microRNA in differentiation initiation, rather than driving a special lineage commitment. Regarding the miR-27a/b family, there is still a discrepancy in the role and regulation of the individual family members. In androgen-responsive PCa cell lines, mainly the induction of miR-27a (located on chromosome 19) is described [84,85,86]. Only in one study an androgen-responsiveness of miR-27b (located on chromosome 9) in endothelial cells was observed [87]. Nonetheless, in hMSCs, it is mainly miR-27b regulating adipogenesis by targeting the key adipogenic transcription factor PPARγ [26,27]. However, there are several reports from mouse pre-adipocytes verifying the assumption that miR-27a exhibits the same role as miR-27b by repressing adipogenic differentiation of mouse 3T3-L1 cells [61,130,131].

A similar situation is given for the miR-29 family, where miR-29b is involved in adipogenesis of human mesenchymal stem cells [28,29], while both miR-29a and miR-29b are androgen-regulated in PCa cell lines, but also in epididymal cells [106,110,111]. When comparing in vitro differentiated osteoblasts, adipocytes, and chondroblasts to their originating mesenchymal stem cells, Laine et al. found miR-124 to be exclusively expressed in adipocytes and to suppress proliferation of hMSCs [35]. Qadir and colleagues concordantly detected that miR-124 exerts a pro-adipogenic effect on 3T3-L1 mouse preadipocytes [132]. In PCa cell lines, the tumor-suppressive miR-124 and the AR formed a positive feedback loop [114]. As one of the most prominent androgen-regulated microRNAs [63,64,65,66,67], miR-125b is still scarcely studied in terms of adipogenesis. However, it was shown that miR-125b is overexpressed during human SGBS preadipocyte differentiation, though experimental overexpression of miR-125b by mimics inhibited the adipogenic differentiation [37].

These observations point towards a strictly balanced regulatory network including miR-125b, which can be shaped via androgen-induced upregulation of this microRNA towards the inhibition of adipogenesis. MiR-137 is induced via androgen stimulation in PCa cell lines, though progressive promoter methylation is detected from normal prostate tissue to androgen-insensitive PCa cell lines [115]. In line with this finding, the overexpression of miR-137 inhibits hADSC proliferation and differentiation [39]. MiR-148a was found to be CREB-induced upon the adipogenic differentiation of hMSCs and promoting hMSC adipogenesis commitment by targeting the Wnt pathway effector Wnt-1 [41]. In the androgen-responsive PCa cell line LNCaP, miR-148a expression is also induced by androgens, resulting in increased cell proliferation [89,118]. The tumor-suppressive miR-204 is downregulated upon androgen stimulation in PCa cell lines [123]. In line with the androgen-mediated inhibition of the adipogenesis, miR-204 is upregulated in differentiating hADSCs and inhibits the activation of the Wnt pathway, therefore, supporting the initiation of differentiation [48]. Lastly, miR-375 was in several studies shown to be sensitive to androgen regulation [54,98,99,100], although in PCa cell lines the AR triggers upregulation of miR-375, while in differentiating hMSCs and SGBS preadipocytes miR-375 is downregulated upon androgen stimulation. However, Chu and colleagues demonstrated the hypermethylation of the miR-375 promoter region in androgen-insensitive cell lines, therefore, giving a possible explanation for this different reaction of miR-375 to androgen stimulation [100]. Furthermore, the upregulation of miR-375 was demonstrated during adipogenesis [54], therefore making this the first AndroMiR to be experimentally revealed in the androgen-mediated inhibition of adipogenesis.

Finally, one has to bear in mind, that in most cases the androgen-regulation of a given microRNA was analyzed and verified in tissues or cell lines completely different from the tissues or cell lines used for the studies on the involvement of this microRNA in adipogenesis. Therefore, the connections drawn in this chapter are literature-based, but in most cases still speculative, and should be experimentally verified for a realistic assessment of the androgen-regulated miRNome in the inhibition of adipogenesis.

## 5. Conclusions

While over 30 microRNAs/microRNA families have been described to be involved in the regulation of adipogenesis of human mesenchymal stem cells and pre-adipocytes, and over 30 microRNAs/microRNA families are known to be regulated in their expression by the androgen receptor, knowledge about the AndroMiRs involved in the regulation of the adipogenesis is still scarce. Therefore, based on this review, further experimental studies on the interplay between the AR-mediated miRNome regulation and the inhibition of adipogenesis are needed and also promising, especially in the light of the growing obesity epidemics and the well-known clinical effects of testosterone treatment on reducing adipose tissue in hypogonadal men [133,134].

## Figures and Tables

**Figure 1 ijms-20-05767-f001:**
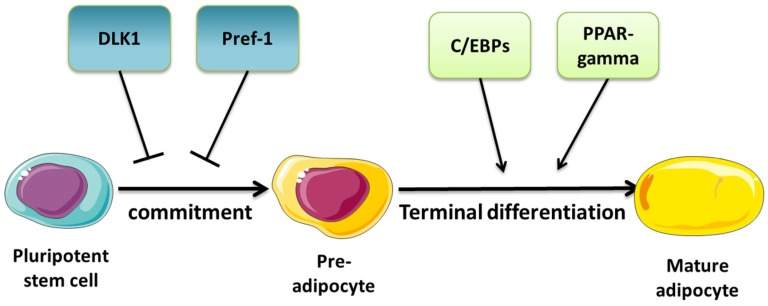
Phases of the adipogenic differentiation. Abbreviations: DLK1—delta-like homolog 1, Pref-1—preadipocyte factor 1.

**Figure 2 ijms-20-05767-f002:**
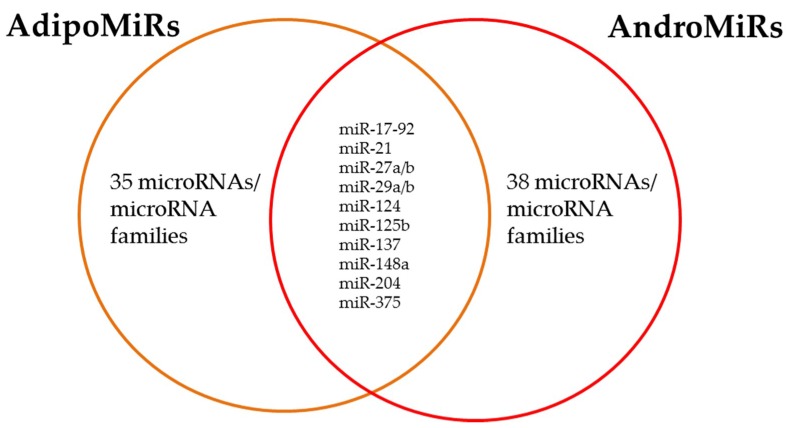
Putative overlap of microRNAs identified as adipogenesis-related or androgen-regulated so far.

**Table 1 ijms-20-05767-t001:** Adipogenesis-regulating microRNAs in human cell lines/preadipocytes.

microRNA	Effect on Adipogenesis	Cell System	Target Gene/ Signaling Pathway	Reference
miR-10b	i	hADSC	CEBPα, PPARγ, AP2,	Li et al., 2018 [21]
mir-17-5p/106a	p	hADSC	BMP2	Li et al., 2013 [22]
miR-21	p	hMSC	SPRY2	Mei et al., 2013 [23]
miR-26b	p	hADSC	PTEN	Song et al., 2014 [24]; Trohatou et al., 2017 [25]
miR-27b	i	hMSC	PPARγ	Karbiener et al., 2009 [26]
miR-27b	i	hADSC	LPL, CEBPα, PPARγ	Hu et al., 2018 [27]
miR-29	i	hMSC	Cyclin D1	Beezhold et al., 2016 [28]
miR-29b	p	hADSC	TNF-α SP-1	Zhang et al., 2016 [29]
miR-30 family	p	hMADS	RUNX2	Zaragosi et al., 2011 [30]
miR-30c	p	hMADS	PAI-1, ALK2	Karbiener et al., 2011 [31]
miR-31	i	hADSC	CEBPα	Liu et al., 2018 [32]
miR-103	p	hADSC	Thy1 (CD90)	Woeller et al., 2017 [33]
miR-107	i	SGBS	CDK6	Ahonen et al., 2019 [34]
miR-124	p	hMSC	FABP4, PPARγ, SOX9	Laine et al., 2012 [35]
miR-125a-3p	p	hADSC	RhoA/ROCK1/ERK1/2	Chen et al., 2015 [36]
miR-125b-5p	p/i	SGBS	MMP11	Rockstroh et al., 2016 [37]
miR-130	i	hMSC	PPARγ	Lee et al., 2011 [38]
miR-137	i	hADSC	CDC42	Shin et al., 2014 [39]
miR-140-5p	i	hMSC	LIFR	Li et al., 2017 [40]
miR-148a	p	hMSC	Wnt1	Shi et al., 2015 [41]
miR-149-3p	i	BMSC	FTO	Li et al., 2019 [42]
miR-155	i	hMSC	PPARγ, CEBPα	Skarn et al., 2012 [43]
miR-192-3p	i	hADSC	SCD, ALDH3H2	Mysore et al., 2016 [44]
miR-193b	p	hADSC	CRKL, FAK	Mazzu et al., 2017 [45]
miR-194	i	hMSC	COUP-TFII	Jeong et al., 2014 [46]
miR-199a-3p	p	BMMSC	KDM6A	Shuai et al., 2019 [47]
miR-204-5p	p	hADSC	DVL3	He et al., 2015 [48]
miR-320 fam.	p	hMSC	RUNX2	Hamam et al., 2014 [49]
miR-335	p	hADSC	MEST	Zhu et al., 2014 [50]
miR-342-3p	p	hMSC	CtBP2	Wang et al., 2015 [51]
miR-363	p	hADSC	E2F3	Chen et al., 2014 [52]
miR-369-5p	i	hMSC	FABP4	Bork et al., 2011 [53]
miR-375	i	SGBS	ADIPOR2	Kraus et al., 2015 [54]
miR-377-3p	i	hMSC	LIFR	Li et al., 2018 [55]
miR-431	i	BMMSC	IRS2	Wang et al.,2018 [56]
miR-483-5p	p	hADSC	RhoA/ROCK1/ERK1/2	Chen et al., 2015 [36]
miR-1908	i	hMADS	PPARγ, CEBPα	Yang et al., 2015 [57]
miR-1275	i	hADSC	ELK1	Pang et al., 2016 [58]
miR-4739	p	hMSC	LRP3	Elsafadi et al., 2017 [59]

Abbreviations: p—promoting effect, i—inhibiting effect, hADSC—human adipose-derived stem cells, hMSC—human mesenchymal stem cell, SGBS—Simpson Golabi Behmel syndrome cells, BMMSC—bone marrow-derived stem cells, hMADS—human multipotent adipose-derived stem cells.

**Table 2 ijms-20-05767-t002:** Androgen-regulated microRNAs (AndroMiRs).

microRNA	Androgen Regulation	Tissue	Cell Line	Target Gene/Signaling Pathway	Reference
let-7a	u	BCA	MCF-7; MDA-MB-231; MDA-MB-453	KRAS; CMYC	Lyu et al., 2014 [101]
let-7c	d	PCA	LNCaP	IGFR1	Sun et al., 2014 [66]
let-7d	u	PCA	LNCaP, C4-2B	PBX3	Ramberg et al., 2011 [102]
miR-1	u	PCA	LNCaP	SRC	Liu et al., 2015 [94]
miR-1	u	PCA	LNCaP	TCF7	Siu et al., 2017 [96]
miR-1	u	PCA	LNCaP	ZBTB46	Chen et al.,2017 [97]
miR-17-92 cluster	u	PCA	LNCaP; 22Rv1	ATG7	Guo et al., 2016 [107]
miR-17-92 cluster	u	PCA	DUCaP; LNCaP	-	Pasqualini et al., 2015 [106]
miR-19a	u	PCA	LNCaP	SUZ12; RAB13; SC4MOL; PSAP; ABCA1	Mo et al., 2013 [85]
miR-21	u	PCA	LNCaP; LAPC-4	-	Ribas et al., 2009 [69]
miR-21	u	PCA	LNCaP	-	Ribas et al., 2010 [70]
miR-21	u	HCC	HepG2	PDCD4	Teng et al., 2014 [72]
miR-21	u	PCA	BPH-1; 22Rv1; PC-3	TGFBR2	Mishra et al., 2014 [71]
miR-21	d	BCA	MCF-7	-	Casaburi et al., 2016 [73]
miR-22	u	placenta	JEG-3	ESR1	Shao et al., 2017 [113]
miR-22	u	PCA	DUCaP; LNCaP	LAMC1	Pasqualini et al., 2015 [106]
miR-23b	d	mouse Sertolicells	-	PTEN	Nicholls et al., 2011 [112]
miR-27a	u	PCA	LNCaP	PHB	Fletcher et al., 2012 [84]
miR-27a	u	PCA	LNCaP	ABCA1; PDS5B	Mo et al., 2013 [85]
miR-27a	u	PCA	LNCaP, 22Rv1; Du145; PC3	MAP2K4	Wan et al., 2016 [86]
miR-27a/b	d	endothelial cell lines	EA.hy926; HUVEC	TFPIα	B Arroyo et al., 2017 [87]
miR-29a	u	PCA	DUCaP; LNCaP	Mcl-1	Pasqualini et al., 2015 [106]
miR-29a/b	d	epididymis	PC-1	AR, IGF1	Ma et al., 2013 [110]
miR-29b	u	PCA	LNCaP; BicR	TET2	Takayama et al., 2015 [111]
miR-30d	d	Sertoli cells	-	-	Nicholls et al., 2011 [112]
miR-32	u	PCA	22Rv1; LNCaP;RWPE1	NSE	Dang et al., 2015 [7]
miR-32	u	PCA	LNCaP	BTG2	Jalava et al., 2012 [89]
miR-99a	d	PCA	LNCaP	IGFR1	Sun et al., 2014 [66]
miR-100	d	BCA	MDA-MB-453	MMP13	Ahram et al., 2017 [67]
miR-101	u	PCA	LNCaP	Ezh2	Cao et al., 2010 [74]
miR-101	u	PCA	LNCaP	-	Guo et al., 2015 [77]
miR-101	u	granulosa cells	primary mouse GCs; KGN	Ezh2	Ma et al., 2017 [75]
miR-101	u	neuronal cells	SH-SY5Y; U251	CYP2D6	Li et al., 2015 [76]
miR-124	u	PCA	PC3; LNCaP	AR	Chu et al., 2015 [114]
miR-125b	u	PCA	LNCaP; cds1	Bak1	Shi et al., 2007 [63]
miR-125b	u	PCA	LNCaP	NCOR2	Yang et al., 2012 [64]
miR-125b	d	PCA	LNCaP	-	Sun et al., 2014 [66]
miR-125b	u	PCA	LNCaP	-	Yang et al., 2015 [65]
miR-125b	d	BCA	MDA-MB-453	MMP13	Ahram et al., 2017 [67]
miR-128-2	u	neuronal cells	SH-SY5Y; U251	CYP2D6	Li et al., 2015 [76]
miR-133b	u	PCA	LNCaP	CDC2L5; PTPRK; RB1CC1; CPNE3	Mo et al., 2013 [85]
miR-133b	u	PCA	LNCaP	-	Yang et al., 2015 [65]
miR-135a	u	PCA	LNCaP; PC-3	ROCK1, ROCK2	Kroiss et al., 2015 [84]
miR-135a	u	PCA	LNCaP	MMP11, RBAK	Wan et al., 2016 [104]
miR-137	u	PCA	LNCaP	KDM2A, KDM4A; KDM5B; KDM7A; MED1	Nilsson et al., 2015 [115]
miR-141	u	PCA	LNCaP, VCaP	-	Waltering et al., 2011 [105]
miR-141	u	PCA	LNCaP	-	Tiryakioglu et al., 2013 [99]
miR-141	u	PCA	tissue	-	Nguyen et al., 2013 [98]
miR-141	u	PCA	LNCaP	-	Gezer et al., 2015 [116]
miR-145	d	RCC	ACHN; SCRC-2; SW-839	HIF2α	Chen et al., 2015 [117]
miR-148a	u	PCA	LNCaP	CAND1	Murata et al., 2011 [118]
miR-148a	u	PCA	LNCaP	-	Jalava et al., 2012 [89]
miR-182-5p	u	PCA	LNCaP	ARRDC3	Yao et al., 2016 [119]
miR-185-5p	u	RCC	SW839	VEGF-c; HIF2α	Huang et al., 2017 [120]
miR-190a	d	PCA	LNCaP	AR; YB1	Xu et al., 2015 [121]
miR-193a-3p	u	PCA	LNCaP; C4-2B	AJUBA	Jia et al., 2017 [122]
miR-200a/b	u	PCA	LNCaP; PC-3; Du145	ZEB2	Jacob et al., 2014 [93]
miR-200a-c	u	PCA	PC-3-AR	-	Williams et al., 2013 [92]
miR-203	u	PCA	LNCaP; 22Rv1	SRC	Siu et al., 2016 [95]
miR-204	d	PCA	LNCaP, 22Rv1	XRN1	Ding et al., 2015 [123]
miR-216a	u	HCC	tissue	TSLC1	Chen et al., 2012 [108]
miR-216a	u	PCA	LNCaP	-	Miyazaki et al., 2015 [109]
miR-221/222	d	PCA	LNCaP; C4-2B	-	Gui et al., 2017 [79]
miR-363	u	BCA	MCF-7	IQWD1	Nakano et al., 2013 [124]
miR-375	u	PCA	LNCaP; C4-2; 22Rv1; PC-3; Du145	-	Chu et al., 2014 [100]
miR-375	d	hMSC	SGBS	ADIPOR2	Kraus et al., 2015 [54]
miR-375	u	PCA	LNCaP	-	Tiryakioglu et al., 2013 [99]
miR-375	u	PCA	tissue	-	Nguyen et al., 2013 [98]
miR-421	d	PCA	LNCaP; 22Rv1; Du145; PC-3	NRAS, PRAME, CUL4B, PFKMB2	Meng et al., 2016 [125]
miR-471-5p	u	Sertoli cells	primary cells	LAP	Panneerdoss et al.,2017 [126,127]
miR-690	d	Sertoli cells	-	-	Nicholls et al., 2011 [112]
miR-2909	u	PCA	LNCaP	TGFBR2	Ayub et al., 2017 [128]
miR-4496	u	PCA	LNCaP	β-Catenin	Wang et al., 2018 [129]

Abbreviations: d—downregulated; u—upregulated; PCA—prostate carcinoma; BCA—breast cancer; HCC—hepatocellular carcinoma; RCC—renal cell carcinoma.

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
