# Peer review of "Androgen-Regulated microRNAs (AndroMiRs) as Novel Players in Adipogenesis"

_ijms, 2019, doi:10.3390/ijms20225767_

Round 1

Reviewer 1 Report

The manuscript submitted by Jansen et al., entitled “Androgen-regulated microRNAs (AndroMiRs) as novel players in adipogenesis” is an excellent review article, which offers novel and updated information regarding microRNAs involved in adipogenesis by modulating androgenic pathways, as well as their implications in the pathophysiology of neoplastic or metabolic diseases.

The Introduction and objectives of the article are appropriately and clearly detailed. The presentation of the data throughout the review, as well as their Discussion, are meaningful and well explained. All the Figures and Tables, as well as the Supplementary Material, support the data. Moreover, the Conclusions summarize adequately the main concepts of such a review.

However, in order to improve the manuscript, only some few minor suggestions should be adequately addressed:

Specific comments:
- Page 2, Line 77: Please, perhaps it would be better if Table 1 appears completely on the same page, including its corresponding legend. Accordingly, please try to place such legend, as well as its table 1, jointly on Page 3.
- Page 4, Line 135: Please, immediately after the word "Furthermore", a comma should be added.
- Page 7 to 9, Table 2: Please, check for indentations on Table 2, and adjust them adequately before the text.

Author Response

We want to thank the reviewer for the positive revision of our manuscript and the thoughtful remarks on content and structure. We have formatted the table and legend adjustment to fit on the same page, if possible. Additionally, commas were added after every “Furthermore”, if applicable (marked in yellow). Lastly, we have re-formatted table 2 to enhance its clarity and readability.

We thank the reviewer for his time and input that we really appreciate. Best regards.

Julia Jansen, Thomas Greither, Hermann M. Behre

Reviewer 2 Report

The manuscript is a review on (i) androgen-regulated microRNAs, (ii) adipogenesis-associated microRNAs, and (iii) the connections/overlaps between these two groups of miRNAs, each consisting of >30 miRNAs/miRNA families. The topic is well motivated, considering that androgen receptor controls the expression of a number of miRNAs, that a large number of miRNAs are known to control adipogenesis, and that hypogonadism induces adipose tissue accumulation, bringing up the possibility of a potential shared miRNA-mediated regulation.

The review is professionally written and its Tables 1 and 2 provide a useful dictionary-like resource for the reader. The content of the Tables is opened in paragraphs 2 and 3, which are quite demanding to read – due to the listing of a large number of miRNA species and reported functional effects. This is an inherent problem of miRNA reviews: many groups have studied different individual miRNAs or miRNA families in various functional assays and cell contexts, which makes it difficult to form a clear synthesis of the published data. Since individual miRNAs regulate tens, often hundreds of distinct genes, and each gene or biochemical pathway is regulated by a large number (at least tens, even hundreds) of miRNA species, generation of coherent and clear functional summaries is an extremely hard task. From this perspective, paragraphs 2 and 3 are quite adequate.

I have no major criticisms concerning the content of this well written review, but rather only minor points to be corrected:

1, line 29: ’mamma’, what is this? I do not know such an English word. P. 2, line 66: ’cross-linking’ sounds strange in this context, since it generally refers to chemical cross-linking. Please replace by ’connecting’ 3, line 82: ’by repressing mothers against…’ I do not understand, what is the meaning of ’mothers’ here? 4, line 112: Past tense ’Knockdown of miR-149-3p led to…’ 4, lines 128-9: ’and the other way around’ It is not clear exactly what this means. Please explain in more detail with a 2nd full sentence. 4, lines 132-4: The sentence ’Mei et al..’ does not work linguistically. The connection with ’whereas’ does not make sense. Please rewrite. 4, line 147:   ’..a key regulator balancing adipogenesis…’ Lines 147-148: Should be ’The miR-320 family has a similar effect [42].’ 5, line 153-4: ’and other way around.’ It is not clear exactly what this means. Please explain in more detail with a 2nd full sentence. Line 154: ’..was confirmed as a direct target….’ 6, line 205: ’In line with these findings….’ 6, line 221: ’..to be upregulated and to enhance NSE activity,…’ 9, line 273: ’….tissues remains crucial for the….’ 10, line 290: ’…adipocytes and to suppress….’ 10, line 299, 309, 322, 327 and 329: Please use OF GENETIVE instead of just placing words after each other; e.g. NOT ’adipogenesis inhibition’ BUT ’inhibition of adipogenesis’ or ’initiation of differentiation’ 10, line 300: ’progressive promotor methylation’ 10, line 309: ’…studies shown to be sensitive to androgen…’ 10, line 317: ’Finally, one has to bear in mind that…’

Author Response

The manuscript is a review on (i) androgen-regulated microRNAs, (ii) adipogenesis-associated microRNAs, and (iii) the connections/overlaps between these two groups of miRNAs, each consisting of >30 miRNAs/miRNA families. The topic is well motivated, considering that androgen receptor controls the expression of a number of miRNAs, that a large number of miRNAs are known to control adipogenesis, and that hypogonadism induces adipose tissue accumulation, bringing up the possibility of a potential shared miRNA-mediated regulation.

The review is professionally written and its Tables 1 and 2 provide a useful dictionary-like resource for the reader. The content of the Tables is opened in paragraphs 2 and 3, which are quite demanding to read – due to the listing of a large number of miRNA species and reported functional effects. This is an inherent problem of miRNA reviews: many groups have studied different individual miRNAs or miRNA families in various functional assays and cell contexts, which makes it difficult to form a clear synthesis of the published data. Since individual miRNAs regulate tens, often hundreds of distinct genes, and each gene or biochemical pathway is regulated by a large number (at least tens, even hundreds) of miRNA species, generation of coherent and clear functional summaries is an extremely hard task. From this perspective, paragraphs 2 and 3 are quite adequate.

First, we want to thank the reviewer for the positive review of our manuscript and the insightful comments. We agree with the reviewer that distilling a meaningful and clearly structured review from the plethora of data on microRNA and their regulated pathways was challenging. We tried to include especially microRNA target genes and regulated pathways, which are involved in differentiation processes, and leaving aside target genes maybe identified in another context. Therefore, we appreciate the reviewer’s positive assessment of paragraph 2 and 3.

I have no major criticisms concerning the content of this well written review, but rather only minor points to be corrected:

Page 1, line 29: ’mamma’, what is this? I do not know such an English word.

We have now replaced the term “mamma” with the more adequate term “breast”.

2, line 66: ’cross-linking’ sounds strange in this context, since it generally refers to chemical cross-linking. Please replace by ’connecting’

We have replaced “cross-linking” or “cross-link/s” with the term “connecting” or “connection/s”

3, line 82: ’by repressing mothers against…’ I do not understand, what is the meaning of ’mothers’ here?

We apologize for the confusion. As we tried to state the full name of every gene abbreviation at the first occurrence, especially longer gene names can be confused with parts of the main text. This seemed to the case in the full name of SMAD4, which is ‘mothers against decapentaplegic homolog 2’. We tried to avoid this type of confusion by adding ‘ at the beginning and end of every gene name longer than two words, hoping that this solution enhances the readability and clarity of the text.

4, lines 128-9: ’and the other way around’ It is not clear exactly what this means. Please explain in more detail with a 2nd full sentence.

5, line 153-4: ’and other way around.’ It is not clear exactly what this means. Please explain in more detail with a 2nd full sentence. Line 154: ’..was confirmed as a direct target….

We agree with the reviewer that the expression “and the other way around” is not well defined and therefore unclear. In short, this expression was chosen to state that by modulating the expression of miR-X in the opposite direction, also the opposite effect on gene Y is exerted than before. As this fact does not add valuable information to the corresponding context, the phrase was omitted from both sentences (see l. 130 and l. 156).

4, lines 132-4: The sentence ’Mei et al..’ does not work linguistically. The connection with ’whereas’ does not make sense. Please rewrite.

We changed the description of the results generated by Mei and colleagues as follows: “Mei et al. demonstrated that through regulating the ERK-MAPK pathway, the only active signaling during adipogenic, osteogenic and chondrogenic differentiation, miR-21 stimulates MSC differentiation on an early stage. In this context, the expression of the marker gene for adipogenesis, PPARγ, and the marker gene for osteogenesis, Cbfa-1, were both increased after transfection of MSC with miR-21 mimics, while miR-21 inhibition resulted in a reduced expression level of both genes [36].” (see line 132 – 137)

4, line 112: Past tense ’Knockdown of miR-149-3p led to…’

4, line 147:   ’..a key regulator balancing adipogenesis…’ Lines 147-148: Should be ’The miR-320 family has a similar effect [42].’

6, line 205: ’In line with these findings….’

6, line 221: ’..to be upregulated and to enhance NSE activity,…’

9, line 273: ’….tissues remains crucial for the….’

10, line 290: ’…adipocytes and to suppress….’

10, line 299, 309, 322, 327 and 329: Please use OF GENETIVE instead of just placing words after each other; e.g. NOT ’adipogenesis inhibition’ BUT ’inhibition of adipogenesis’ or ’initiation of differentiation’

10, line 300: ’progressive promotor methylation’

10, line 309: ’…studies shown to be sensitive to androgen…’

10, line 317: ’Finally, one has to bear in mind that…’

We fixed these orthographic or syntactic mistakes accordingly.

Finally, we thank the reviewer for his time and input that we really appreciate. Best regards.

Julia Jansen, Thomas Greither, Hermann M. Behre